# Requests for futile treatments: what mechanisms play a role? Results of a qualitative study among Dutch physicians

Rozemarijn Lidewij van Bruchem-Visser  ,[1] Gert van Dijk,[2] Francesco Mattace Raso,[3] Inez de Beaufort[2]

[1]Internal Medicine, Erasmus Medical Center, Rotterdam, The Netherlands
[2]Medical Ethics and Philosophy of Medicine, Erasmus Medical Center, Rotterdam, Zuid-Holland, The Netherlands
[3]Section of Geriatric Medicine, Department of Internal Medicine, Erasmus MC, Rotterdam, Zuid-Holland, The Netherlands

**Correspondence to**
Rozemarijn Lidewij van Bruchem-Visser;
r.l.visser@erasmusmc.nl

## ABSTRACT

**Objectives** Overtreatment is increasingly seen as a challenge in clinical practice and can lead to unnecessary interventions, poor healthcare outcomes and increasing costs. However, little is known as to what exactly causes overtreatment. In 2015, the Royal Dutch Medical Association (RDMA) attempted to address this problem and distinguished several mechanisms that were thought to drive overtreatment. In 14 qualitative interviews among Dutch physicians, we investigated which mechanisms played a role in decision-making and whether all mechanisms were considered equally important.

**Design** We asked physicians to present a case from personal experience, in which the patient or family requested continuing treatment against the advice of the physician.

**Participants** Fourteen physicians from five different medical areas agreed to participate.

**Setting** Interviews were held face-to-face at the workplace of the physician.

**Results** Three closely related mechanisms were mentioned most frequently as drivers of overtreatment, as perceived by the physician: 'death is not a common topic of conversation', "never give up' is the default attitude in our society' and 'patients' culture and outlook on life influences their perception of death'. The mechanism 'medical view taking priority' was mentioned to be an inhibitor of overtreatment.

**Conclusions** Of the 15 mechanisms described by the report of the Steering Committee of the RDMA, not all mechanisms were mentioned as driving overtreatment. Three mechanisms were mentioned most as being a driver of overtreatment ('death is not a common topic of conversation'; "never give up' is the default attitude in our society'' and 'patients' culture and outlook on life influences their perception of death'), some played no role at all, and others were considered to be inhibitors of overtreatment, especially the mechanism 'medical view taking priority'.

## INTRODUCTION

Overtreatment is increasingly seen as a challenge in clinical practice and can lead to unnecessary interventions, poor healthcare outcomes, and increasing costs.[1] The

### Strengths and limitations of this study

► Physicians from different specialties and working experiences were interviewed, making the findings more generalisable.
► A wide variety of patients facing end-of-life decisions was described, making the findings applicable to several types of situations and/or physicians.
► All interviewed physicians work in The Netherlands.
► Opinions on what futile treatment is, might differ between different countries or cultures.

occurrence of overtreatment is acknowledged by both patients and physicians in all patients groups, including the elderly.[2–4]

'Overtreatment' or 'too much medicine' may result from overdiagnosis, in which occurs people are 'labelled with or treated for a disease that would never cause them harm'.[5 6]

Overtreatment includes: 'treatments that are unnecessary or inappropriate'; 'unnecessary investigations and treatment that lack patient benefit or bear the potential to cause harm'[7]; treatment that is not 'in line with patient's wishes'; 'the provision of medical services for which the potential for harm exceeds the potential for benefit'[8] or 'treatment initiated when there is little or no reliable evidence of a clinically meaningful net benefit, where net benefit equals benefit minus harm'.[9] Overtreatment can concern interventions that benefit specific patient groups, but can harm others. For example, percutaneous endoscopic gastrostomy (PEG) tubes can be very beneficiary in some categories of patients, but will harm patients with advanced dementia.[10]

### Subjective component

In applying the definition of overtreatment, the patient plays a crucial role. The

assessment that a certain intervention is overtreatment in a specific situation can differ between patients, family, and physicians. This can make it difficult to objectively determine whether overtreatment has occurred in a specific situation. For instance, what has 'benefit' for a patient can often not objectively be determined as patient preferences differ. Some patients will request as much treatment as possible and accept the smallest chance of success among substantial risk, while other patients will be much more reluctant towards interventions. Medical interventions can also have other benefits; an intervention that has little chance of medical success, but provides the feeling that every option has been explored, can still be seen as 'useful' by the patient.

### Foster appropriate care

The Royal Dutch Medical Association (RDMA) established the Steering Committee for Appropriate End-of-Life Care (SCoAEoLC) to address the problem of overtreatment and to 'foster appropriate care for those nearing the end of life'. One of the main tasks of the steering group was to identify mechanisms that are thought to drive overtreatment. In 2015, theSCoAEoLC published a report "Just because we can, doesn't mean we should, appropriate end-of-life care".[11] The report investigated several mechanisms thought to drive overtreatment. These mechanisms play a role in several different domains: society in general, the healthcare system, industry, professionals, patients, and the public. These mechanisms are in accordance with similar findings from a recent study in Germany.[7]

Table 1 shows the 15 mechanisms that were deemed to be the most important drivers of overtreatment.

The aim of the present study was to determine which of the mechanisms described by the SCoAEoLC were recognised by Dutch physicians of different medical specialties to play a role in driving overtreatment. As the SCoAEoLC has primarily focused on patients in the last phase of life, defined as 'being of old age or suffering from a terminal disease with limited life expectancy', we wondered whether these mechanisms do play a role in the clinical practice of Dutch physicians.

## METHODS

### Study setting and population

We purposively sampled 18 physicians differing in years of working experience and medical specialty to participate. Using the accessibility guide of the Erasmus MC and the list of general practitioners in the region, names were randomly selected from the different departments of (academic) hospitals or groups of local general practitioners. Fourteen physicians from five different medical areas (internal medicine, general practice (GP), intensive care, surgery and oncology) agreed to participate. Four physicians were interested in the topic, but were not able to find sufficient time in their schedule to participate. The work experience of participating physicians ranged from 1 to 35 years. All interviews were conducted between March 2014 and November 2015.

A medical doctor (RLvB-V, internist geriatrician) conducted semi-structured interviews with all physicians, face-to-face at the workplace of the physician. Before the interview, participants were informed that the interview would deal with 'disagreement between physician and patient regarding the course of treatment'. Physicians were asked to recall a case from their own personal experience, in which the patient or family requested continuing or starting a treatment that was not offered or advised by the physician. No limitations were given regarding type of underlying disease, patients' age, sex or cultural background. After narrating the patient's story, the physician was presentend with the fifteen mechanisms described by the SCoAEoLC (table 1) and indicated whether the mechanisms played a role in that case. Participants were given written and verbal information and gave permission to record the interview and store the recordings in a safe location for verbatim transcription.

### Data collection and analysis

Interviews were digitally recorded and transcribed verbatim with the interviewees' permission. Basic demographic information of the described patients (age, sex, ethnicity, religion, relationship with physician) and the physicians (sex, medical specialty, years of working experience) were recorded in tables 2 and 3. The study was facilitated by QSR NVivo 12 software (QRS International, Melbourne, Victoria, Australia).

| Table 1 | Mechanisms described by the Steering Committee for Appropriate End-of-Life Care |
|---|---|
| 1 | Death is not a common topic of conversation |
| 2 | 'Never give up' is the default attitude in our society |
| 3 | Action is better than inaction |
| 4 | Professional guidelines focus on 'action' |
| 5 | Education focuses on 'action' |
| 6 | Physicians are payed for treatment |
| 7 | With so many care providers and so little coordination, who is responsible? |
| 8 | No holistic view of the patient |
| 9 | Medical perspectives often still take priority when it comes to making treatment decisions |
| 10 | Palliative care is initiated too late |
| 11 | Discussing possible refusal of treatment is more time-consuming |
| 12 | To talk about death is difficult |
| 13 | Uncertainty about what to tell patients |
| 14 | The great unknown: patients' culture and outlook on life influences their perception of death |
| 15 | People document their wishes and preferences regarding end-of-life care too late, and often not (thorough enough) |

**Table 2** Descriptive characteristics of patients

| Total number of patients | 14 |
| --- | --- |
| Age (years) | |
| 25–45 | 6 |
| 46–65 | 3 |
| 65 years or older | 5 |
| Sex | |
| Male | 4 |
| Female | 10 |
| Underlying disease | |
| Malignancy | 5 |
| Kidney disease | 2 |
| Diabetes mellitus | 0 |
| Neurological condition | 4 |
| Chronic obstructive pulmonary disease | 0 |
| Infection | 0 |
| Complex surgery | 1 |
| Medically unexplained | 1 |
| Dementia | 1 |
| Religious background | |
| Non-religious/not mentioned | 4 |
| Muslim | 5 |
| Christian | 2 |
| Other | 3 |
| Existing relationship patient-physician | 7 |
| Treatment requested by patient | 5 |
| Treatment requested by one relative | 3 |
| Treatment requested by more than one relative | 6 |
| Mentally incompetent | 6 |
| End-of-life situation | 14 |

The answers to the mechanisms were coded by two independent researchers (RLvB-V and GvD) to ensure rigorous analysis and to assess whether that mechanism was a factor in the described case. Disagreements were

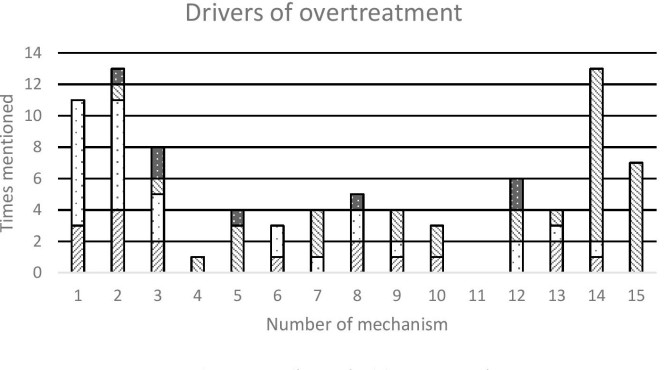

**Figure 1** Drivers of overtreatment, attributed to different parties.

**Table 3** Descriptive characteristics of the interviewed physicians

| Participant | Sex | Specialty | Working experience (years) |
| --- | --- | --- | --- |
| 1 | Female | General practitioner | 22 |
| 2 | Male | Intensive care | 10 |
| 3 | Female | Oncology | 11 |
| 4 | Male | Surgery | 30 |
| 5 | Female | Intensive care | 35 |
| 6 | Male | Internal medicine | 4 |
| 7 | Female | Oncology | 20 |
| 8 | Female | General practitioner | 11 |
| 9 | Female | Internal medicine | 1 |
| 10 | Female | Internal medicine | 15 |
| 11 | Male | Surgery | 1 |
| 12 | Male | Surgery | 10 |
| 13 | Female | Oncology | 2 |
| 14 | Female | Intensive care | 6 |
| 15 | Female | General practitioner | 3 |

settled by consensus. The coding tree was based on the interview guide of the 15 mechanisms. We used the SRQR checklist when writing our report.[12]

### Patient and public involvement

Patients or the public were not involved in the design, conduct or reporting of our research.

### RESULTS

All physicians described a patient facing an end-of-life situation. In five patients' stories, the patient was the one requesting a treatment. In the other nine cases, the relatives (surrogate decision-makers such as spouses or children) were requesting further treatment. Six patients were 25-45 years old, three were 45-65 years old and five were 65 years or older. The physicians chose their patient stories for varying reasons, either the young age of the patient, the acuteness of the situation or the frustration when a requested treatment was felt to be unnecessary, futile or even harmful to the patient.

Fourteen different mechanisms were mentioned to play a role in overtreatment, in total 103 times. Mechanisms were identified as either drivers of overtreatment (14 different mechanisms, in total 86 times, see figure 1), or inhibitors of overtreatment (four different mechanisms, in total 17 times, see figure 2). Only mechanism no. 11 'discussing possible refusal of treatment is more time-consuming' was not mentioned as a factor. All the physicians recognized time constraints as a common issue, bit for the described conversations, they devoted all the time necessary.

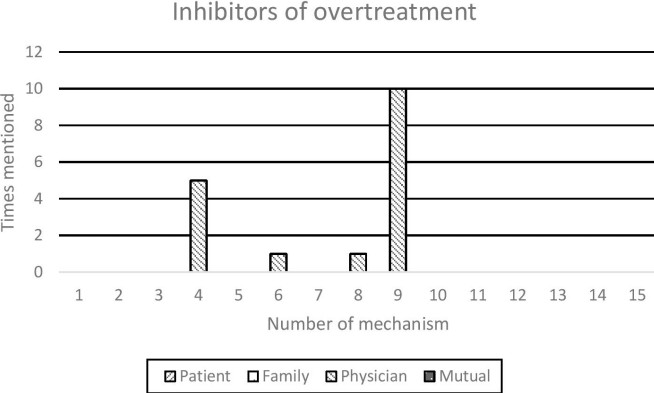

**Inhibitors of overtreatment**

**Figure 2** Inhibitors of overtreatment, attributed to different parties.

We found that according to the interviewed physicians, three closely related mechanisms were considered to be the main drivers of overtreatment: no. 1: 'death is not a common topic of conversation'; no. 2: 'never give up' is the default attitude in our society and no. 14: 'the great unknown: patients' culture and outlook on life influences their perception of death'. These three mechanisms were mentioned 37 times as being drivers of overtreatment.

When attributing the perceived drivers of overtreatment to the different parties, there was a distinct difference between the mechanisms. As is shown in figure 1, mechanism no. 1 'death is not a common topic of conversation' was in all cases attributed to the patient or family, whereas mechanism no. 14 'the great unknown: patients' culture and outlook on life influences their perception of death' was assigned to the physicians themselves. Mechanism no. 2 'never give up' is the default attitude in our society was mainly attributed to patients and family, with two cases assigning this mechanism to the physician or a combination of parties.

Of the mechanisms seen as potential inhibitors, mechanism nine: 'medical perspectives often still take priority when it comes to making treatment decisions', was most frequently mentioned (10 times). As shown in figure 2, this mechanism was in all cases attributed to the physician.

In 11 cases, the physician was not able to communicate directly with the patient, either because the patient spoke a different language and the physician had to communicate by way of the relatives (5 cases) or because the patient was incapacitated (6 cases).

### Death is not a common topic of conversation
The tendency to 'not give up' was considered to be a factor in 13 out of the 14 described cases. The difficulties of talking about death and end-of-life topics in general, were in most cases attributed to the patient and family.

The problem was that, he (the husband of the patient) did not want to talk about his wife dying, because that was just not going to happen. (Interview 8)

The family was not used to talk about the end of life, no. But they found it also hard to talk about bad news in general. (Interview 10)

It is sometimes thought physicians themselves find it difficult to talk about death, but this is not what we found in our interviews. Physicians described these kind of conversations as being part of their normal daily routine.

Actually, we talked about it over and over again. (Interview 9)

Only because she refused to talk about it, not because we were not willing to discuss the subject. (Interview 15)

However, two physicians described their own struggles with talking about death. In both cases, their hope to cure the patient was the cause of reluctance to talk about a bad outcome.

So, in this case, while I do possess the skills to talk about it, I found it extremely difficult. (Interview 8)

It caused frustration on my part, and also a feeling of helplessness. (Interview 13)

### 'Never give up' is the default attitude in our society
In our interviews, we found this mechanism was recognised as being an important factor in almost all cases. In 13 cases, physicians described that the patient or family did not want to give up, even when the physician had told them further treatment would be futile.

This family, most definitely. They wanted us to pull out all the stops, give all possible treatments. (Interview 10)

The tendency to not give up, is not something that always arises from the patient, but can sometimes be pushed by relatives, not from a wish to harm the patient, but out of love and empathy:

Yes. It does play a role. They would rather have her be subjected to a dozen futile treatments than… They were not trying to make memories with their loved one, no, they were still searching the internet for some treatment the doctor had overlooked. (Interview 12)

In cases where there was an acute illness, with little time to decide and often little knowledge about the patient due to the acuteness of the situation, physicians described they automatically opted to initially start treatment. To end a treatment, or to not continue on a direction of treatment was described to be difficult.

When you get a patient with acute renal failure, we have no time to think about other options. To act is the default position. (Interview 6)

Yes, that did play a role. Especially with the patient, but also with my supervisor, at the start of this process. He was involved in the first part of this case, before I took over. He had been compliant to her wishes so

far, had suggested and arranged the percutaneous endoscopic gastrostomy. So I do think this played a role in the first stage of this case. (Interview 15)

### The great unknown: patients' culture and outlook on life influences their perception of death

Physicians described, sometimes in detail, their unfamiliarity with cultural differences between themselves and (family/relatives of) their patients.

On their part, it was most certainly difficult. Because of religion and culture. (Interview 5)

We found out too late that this daughter had never told her mother the diagnosis, because in their culture, younger family members were not allowed to convey bad news to their elders. (Interview 10)

### Medical perspectives often still take priority when it comes to making treatment decisions

In our interviews, physicians described using the medical perspective to convince patients not to continue treatment.

Well yes. I told them I would not perform a CAT scan because there was no medical reason to do so. So I use the medical perspective to explain why I withhold certain interventions. (Interview 7)

In this case the medical perspective led to our decision to tell the patient 'enough is enough'. Locally, we would be able to do a lot of things, but the fact the same problem would come back in different places, or that the wound would never close, led to us saying 'we will not give any further treatment. (Interview 10)

Taking a medical perspective can have varying effects. When the physician places too much focus on a specific organ or biological explanation, other factors can be overlooked. However, a medical perspective can also serve as an argument to discontinue medically futile treatment.

## DISCUSSION

Of the fifteen mechanisms described by the SCoAEoLC, not all mechanisms were considered important in driving overtreatment. Three mechanisms were mentioned most as being the drivers of overtreatment ('death is not a common topic of conversation', "never give up' is the default attitude in our society' and 'patients' culture and outlook on life influences their perception of death'), some played no role at all and others were considered to be inhibitors of overtreatment, especially the mechanism 'medical view taking priority'.

The three mechanisms that were mentioned most as contributing to overtreatment are closely intertwined.

Patients are often hesitant to discuss the approach to end of life and to accept the fact that they are going to die. This reluctance to talk about the end of life is sometimes enhanced by relatives, who can pressure patients not to give up.[13] It is difficult to talk about death when the patient and family are still aiming for a cure. Furthermore, in certain cultures it is not customary to discuss the diagnosis, or approaching death with the patient, making it even harder for the physician to determine what is appropriate care in specific cases. These factors interfere with open communication in the acute setting of an end-of-life situation, often leaving patients' preferences and values unknown to their physicians. This was the case in the majority of the narrated patient stories.

The general tendency in society is 'to not give up' when diagnosed with a disease. Dealing with a disease is seen as a 'fight' or a 'war'.[14] If patients are cured they are seen as 'winners' who have 'conquered' the disease. These warlike metaphors are also seen in funding campaigns, such as 'the war on cancer'.

The cultural and spiritual background of patients can be contributing factors in the perception of physicians that the patient is being subjected to overtreatment. For instance, in some cultures it is not common practice to inform the patients of diagnosis and prognosis. Patients and relatives can also hold specific views on the meaning of suffering and whether pain medication is indicated: 'suffering is purification'. These views can put patients in a situation that physicians find difficult to accept professionally.

Due to study design, overtreatment was determined only according to the physician's perspective. It remains unclear whether the patients or relatives also considered the proposed intervention to be a case of overtreatment. This would require further research. The question remains therefore, whether physicians and relatives have similar views on whether a certain intervention is to be considered overtreatment.

In their report, the SCoAEoLC investigated these mechanisms through literature studies, consultations with experts from a range of disciplines and independent research. However, the report was a consensus document and a wider evidence base is required. The report focused on patients in the last phase of life and it remains unknown whether these same mechanisms will apply to other cases of perceived overtreatment.

Interestingly, while the SCoAEoLC found the medical perspective to be a driver of overtreatment, our interviews indicate medicalization can also be utilized as an inhibitor. According to the SCoAEoLC, physicians may focus too much on the medical aspects of a disease, overlooking other important factors such as the wellbeing of the patient or social and cultural aspects. The medical perspective often dominates the decision-making process, even when multidisciplinary consultation is involved. Such bodies often consist only of medical specialists, which can overshadow reflection on social, mental, spiritual, cultural and ideological aspects, as well as general well-being'.[11] However, in our interviews, we found this driver needs to be interpreted in a more nuanced way. In our findings, the 'medical view' was used by physicians as an argument to refuse to provide the intervention under discussion, as the intervention was deemed medically

futile by the physician. This may be attributed to the fact that we interviewed physicians, who may have difficulty recognizing the role this mechanism plays in their relationship with a patient.

This may be atrributed to the fact that we interviewed physicians, who may have difficulty recognising the role this mechanism plays in their relationship with a patient.

Another important finding was that, when asked for a case from their own experience, physicians often presented cases in which the patient was not able to communicate with the physician by herself/himself, either due to a language barrier or mental incapacity. In these cases, it was therefore the relatives and not the patient who opposed the advice of the physicians. The inability to know the wishes of the patient amplified the difficulties already inherent in end-of-life discussions.

One of the reasons for the fact that physicians presented a case in which the relatives played a crucial role might be the subjective aspect of overtreatment. As overtreatment is, at least partly, a subjective phenomenon, physicians might not consider an intervention to be overtreatment when the patient has specifically asked for it. Patients can have different views on what is appropriate care towards the end of life. Physicians are likely to be willing to go along with the patient's wishes, within limits, assuming the patient is well-informed of the associated risks and benefits.

However, when relatives go against the advice of the physician, physicians may have more reservations. Without direct consent from the patient, l the intervention may not serve the medical interest of the patient and is therefore more likely to be considered overtreatment by the physician. A refusal of pain medication or sedatives, or a request for a certain intervention will probably be judged in a different manner by the physician if it comes from the patient as opposed to a request from relatives. Although relatives will almost certainly have the best intentions—"do not let my loved one die"—these intentions might have negative consequences for the patient. It might be acceptable for a physician to see a patient suffer when that patient has deliberately accepted negative consequences of a certain intervention. However, in a mentally incapacitated patient, suffering because of medically futile treatments of any kind is more difficult to accept.

The important role of relatives is not mentioned by the SCoAEoLC as a distinct mechanism that drives overtreatment. This could be explained by the fact that the role of relatives was not a subject of their research. We suggest however, that future research should focus more on the role of relatives in the debate on overtreatment, and the reasons they have for continuation or start of treatments that are deemed inappropriate by the physician. For instance, it would be of interest to find out whether the relatives in these situations considered the intervention to be a case of overtreatment as well, or whether they considered the situation to be appropriate care.

All physicians described a case that had an impact on them. The described patients were generally younger individuals and the decisions to be made were related to end-of-life decision making. Interestingly, (younger) age is not described as a mechanism of driving overtreatment. However, it was mentioned by several physicians that age played a role in the family's view that everything should be done to save the patient's life.

## CONCLUSIONS

We have found that three mechanisms were identified most often as a driving factor in overtreatment: 'death is not a common topic of conversation', "never give up' is the default attitude in our society' and 'the great unknown: patients' culture and outlook on life influences their perception of death'.

The mechanism 'medical perspectives often still take priority when it comes to making treatment decisions' was mentioned as being an inhibitor of overtreatment in the majority of interviews.

In many cases, the relatives play a crucial role in the wish to continue treatment against the advice of the physician. Overtreatment was defined from the perspective of the physicians. Further research is needed to investigate whether relatives of the patient define the same situations as 'overtreatment'.

The findings from this study underline the importance of advance care planning and a timely discussion on patients' wishes and preferences regarding the end of life, so as to avoid overtreatment and to foster appropriate care.

**Acknowledgements** The authors would like to thank all participating physicians.

**Contributors** RLvB-V has made substantial contributions to the conception or design of the work. She has acquired and interpreted the data for the work. She was involved in drafting the work, approved the final version to be published and agrees to be accountable for all aspects of the work in ensuring that questions related to the accuracy or integrity of any part of the work are appropriately investigated and resolved. GvD has made substantial contributions to the conception or design of the work. He was involved in drafting the work, approved the final version to be published and agrees to be accountable for all aspects of the work in ensuring that questions related to the accuracy or integrity of any part of the work are appropriately investigated and resolved. FMR has made substantial contributions to the conception or design of the work. He was involved in revising the work critically for important intellectual content, approved the final version to be published and agrees to be accountable for all aspects of the work in ensuring that questions related to the accuracy or integrity of any part of the work are appropriately investigated and resolved. IdB has made substantial contributions to the conception or design of the work. She was involved in revising the work critically for important intellectual content, approved the final version to be published and agrees to be accountable for all aspects of the work in ensuring that questions related to the accuracy or integrity of any part of the work are appropriately investigated and resolved.

**Funding** The authors have not declared a specific grant for this research from any funding agency in the public, commercial or not-for-profit sectors.

**Competing interests** None declared.

**Patient and public involvement** Patients and/or the public were not involved in the design, conduct, reporting or dissemination plans of this research.

**Patient consent for publication** Not required.

**Ethics approval** The ethical committee of the Erasmus MC was consulted who confirmed that this study is not subject to the Medical Research Involving Human Subjects act.

**Provenance and peer review**  Not commissioned; externally peer reviewed.

**Data availability statement**  Data are available on reasonable request. Data are stored on a bit locked computer in the Erasmus Medical Center, Rotterdam.

**ORCID iD**
Rozemarijn Lidewij van Bruchem-Visser http://orcid.org/0000-0002-6833-7641

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
