## [Reviewer comments · BMJ Open]

ARTICLE DETAILS

TITLE (PROVISIONAL)	REQUESTS FOR FUTILE TREATMENTS: WHAT MECHANISMS PLAY A ROLE? RESULTS OF A QUALITATIVE STUDY AMONG DUTCH PHYSICIANS
AUTHORS	van Bruchem-Visser, Rozemarijn Lidewij; van Dijk, Gert; Mattace Raso, Francesco; de Beaufort, Inez

VERSION 1 – REVIEW

REVIEWER	Sanket Dhruva UCSF School of Medicine, United States
REVIEW RETURNED	28-Dec-2019

GENERAL COMMENTS	This manuscript describes the results of a qualitative (semi-structured interview) study of 14 Dutch physicians to determine which of the drivers of overtreatment identified in a 2015 document (“Just because we can, doesn’t mean we should”) were felt to be pertinent when asking the physicians to describe a case in which patients and/or family members requested treatment against physician advice. Because of the growing recognition of overtreatment and the need to identify barriers and enablers to targeting overtreatment, this study is timely and has the potential to be important. There are several methodological issues and limitations that require additional clarification. First, the authors state in the methods that they identified 18 physicians but only report the results of 14 surveys. Presumably 4 physicians declined to participate. If so, were the reasons for lack of (or declining) participation stated? Additionally, why was ethics approval not required if informed consent was obtained and the interviews were recorded? Second, this study is notable because the physicians were asked to present a specific case from personal experience. If respondents were only asked to present a single case (as it appears), then this is likely to be the most memorable or notable case of overtreatment – perhaps one where the patient was at the end-of-life because the request was to continue treatment against the physician’s advice. This means that the lessons are unlikely to be applicable to the overtreatment that may occur in more routine clinical practice (e.g. use of dietary supplements where benefits are unlikely to outweigh risks or provision of antibiotics for likely upper respiratory tract infections). This also means that there are implications for the mechanisms that are identified – for example, death not being a common topic of conversation is likely because a single case may be most salient when a patient is near-death
--

(as opposed to, for example, another more routine example) and could also be related to the mean age of the patients (see final line of this paragraph). This makes the findings slightly more aligned with the 2015 report from the Steering Committee for Appropriate End-of-Life Care, since that report focused primarily on patients in the last phase of their life. But, again, this leads the findings to be much more circumscribed. To help address this limitation, it would be helpful at least for the authors to report the proportion of patient cases, as reported by physicians, who were felt to be near the end of their life. Of note, the proportion of patients who were younger (age 25-45, 6/14=42.8%, is significantly higher than the average life expectancy – which may have posed additional challenges in addressing overtreatment).

To this end, the title of the manuscript should also be revised to clarify that the manuscript refers to a situation where patients/family requested treatment vs. the physician's advice (i.e. it is not referring to any overtreatment in general).

Also, it is perhaps confusing as to why the treatment was continued against the advice of the physician; if the benefits of treatment were not felt to outweigh the risks, then why were the treatments offered? Can the authors offer at least one-two examples? (e.g. presumably something like continuing ventilatory support or tube feeds despite likelihood of benefit?)

The authors explain that some of the key findings were related to challenges that physicians felt when addressing patients' relatives. This suggests that it may have been superior to stratify results by those when physicians felt overtreatment was requested directly by a patient and when overtreatment was requested by a relative(s). Further, it may be helpful to at least understand the cases in which a larger group of relatives were involved, or to distinguish between cases where a relative was a designated health care proxy / surrogate decision-maker.

Can the authors provide the semi-structured interview templates for review? Specifically, were there specific aspects of the case that physicians were asked to recollect and discuss during the semi-structured interview?

The authors make several important points, and which ideally would be referenced with citations. For example, physicians finding it difficult to talk about death (page 10 of 26, line 19). Some of these points (including page 10 of 26, line 19) would likely fit better in the Discussion (e.g. page 11 of 26, lines 3-8; page 12 of 26, line 11).

The authors state in the first sentence of the Discussion (page 14 of 26) that "not all mechanisms were considered equally important." However, were they evaluating the importance of the mechanisms, more their frequency of mention as well? How were physicians asked to rank the importance of these mechanisms? Similarly, in the conclusions (page 17 of 26, line 8) the authors refer to "the major driving factors" but it is not clear exactly what "major" refers to and the methods should clarify this topic.

Throughout the paper (including in the Title), it may be more accurate to refer to citation 11 by its name instead of only referring to the last name of the first author/Chair of the Steering Committee

	Page 2 of 26, line 30: “addressed this problem” should be “attempted to address this problem” Page 3 of 26: Not all of the items in the Conclusions are mentioned in the Results of the abstract Page 6 of 26, line 55: It is not just that some patients will accept even a small chance of success, but also be willing to accept risks (i.e. medical decision-making is often based on a balance of benefits and risks – not solely on the possibility of benefit) Page 11 of 26, line 54: It is not clear what aspect of this quotation from interview 15.1 relates to “easy to cure” problems. Page 14 of 26, lines 34-39: Is it that patient preferences and values are unknown as the authors state, or is it that they are not elicited by physicians? It seems that the latter is also important Lines 44-46: Is it that the patients or relatives considered the proposed intervention to be overtreatment before the decision or afterwards? Lines 48-51: The question that the authors pose here likely varies across clinical contexts. Additionally, the physicians may consider an intervention to be overtreatment – but another question remains as to why the physicians are offering the intervention to patients. If it is truly overtreatment and benefits do not outweigh risks (or meets one of the other definitions stated in the Introduction’s paragraph starting at line 30), this should have been further explored. While the authors allude to a potential explanation on page 15 (if the patient asks for an intervention then physicians may be most likely to go with it), it is still likely that physicians would have introduced the potential intervention to the patient in most cases. Pages 21 and 22 of 26: The results of Figures 1 and 2 should be discussed in further detail in the paper
--	---

REVIEWER	Loai Albarqouni Bond University, Australia
REVIEW RETURNED	06-Jan-2020

GENERAL COMMENTS	Thanks for inviting me to review this manuscript. The authors in this study aimed to investigate the drivers of overtreatment among 14 Dutch physicians and found that 3 mechanisms played an important role as the main drivers of overtreatment (‘death is not a common topic of conversation’, ‘never give up’ is the default attitude in our society’ and ‘patients’ culture and outlook on life influences their perception of death’). The manuscript is well written and easy to follow. Few comments to strengthen the manuscript: Introduction:  - Need to be revised to make it clear, concise, and to the point connecting the rationale for this research project to the objectives. - Van der Wal mechanisms are not very common, therefore, I wonder whether to include the name in the title? Methods  - Selection/sampling process need to be adequately reported. For instance, author mentioned the 14 were randomly selected by without further details.
---

VERSION 1 – AUTHOR RESPONSE

Reviewers' Comments to Author:

Reviewer: 1

Reviewer Name: Sanket Dhruva

Institution and Country: UCSF School of Medicine, United States Please state any competing interests or state 'None declared': None declared

This manuscript describes the results of a qualitative (semi-structured interview) study of 14 Dutch physicians to determine which of the drivers of overtreatment identified in a 2015 document (“Just because we can, doesn’t mean we should”) were felt to be pertinent when asking the physicians to describe a case in which patients and/or family members requested treatment against physician advice. Because of the growing recognition of overtreatment and the need to identify barriers and enablers to targeting overtreatment, this study is timely and has the potential to be important. We would like to thank the reviewer for his valuable time and effort and the elaborate suggestions and comments. We have revised our manuscript accordingly and feel the content has improved.

There are several methodological issues and limitations that require additional clarification.

First, the authors state in the methods that they identified 18 physicians but only report the results of 14 surveys. Presumably 4 physicians declined to participate. If so, were the reasons for lack of (or declining) participation stated?

This is a valid point and not clarified enough in the original manuscript. We have added a sentence to the methods section:

Methods (page 8, lines 2-3): *Four physicians were interested in the topic, but were not able to find sufficient time in their schedule to participate.*

Additionally, why was ethics approval not required if informed consent was obtained and the interviews were recorded?

As this subject was also mentioned by the editor, it is obvious this topic needs further explanation. At the time the physicians were asked to participate, this category of research was exempted from approval. However, to assure the proper procedures were followed, ethical approval was requested and granted by the Medical Ethical Committee of the Erasmus MC.

The article summary and method section were altered accordingly.

Summary (page 5, line 13): *Ethical approval was granted by the ethical committee of the Erasmus MC.*

Methods (page 8, lines 15-16): *Participants were given written and verbal information and gave permission to record the interview and store them in a safe location, to be used for verbatim transcription. Ethical approval was granted by the ethical committee of the Erasmus MC.*

Second, this study is notable because the physicians were asked to present a specific case from personal experience. If respondents were only asked to present a single case (as it appears), then this is likely to be the most memorable or notable case of overtreatment – perhaps one where the patient was at the end-of-life because the request was to continue treatment against the physician’s advice. This means that the lessons are unlikely to be applicable to the overtreatment that may occur in more routine clinical practice (e.g. use of dietary supplements where benefits are unlikely to outweigh risks or provision of antibiotics for likely upper respiratory tract infections). This also means that there are implications for the mechanisms that are identified – for example, death not being a common topic of conversation is likely because a single case may be most salient when a patient is near-death (as opposed to, for example, another more routine example) and could also be related to the mean age of the patients (see final line of this paragraph). This makes the findings slightly more aligned with the 2015 report from the Steering Committee for Appropriate End-of-Life Care, since that report focused primarily on patients in the last phase of their life. But, again, this leads the findings to be much more circumscribed. To help address this limitation, it would be helpful at least for the authors to report the proportion of patient cases, as reported by physicians, who were felt to be near the end of their life. Of note, the proportion of patients who were younger (age 25-45, 6/14=42.8%, is significantly higher than the average life expectancy – which may have posed additional challenges in addressing overtreatment).

This is a very important comment. The reviewer is right that physicians without exception chose a patient story that had made an impact on them, because of the relative young age of the patient, the acuteness of the situation or the frustration they felt when a treatment was requested or even demanded they had not proposed and felt was unnecessary, futile or even harming the patient. The majority of the cases (13 out of 14) was indeed about an end-of-life situation. We have added these numbers to the result section as well as table 2. Also, a section is added to the discussion section regarding these findings.

Results (page 10, lines 2-8): *All physicians described a patient facing an end-of-life situation. In five patient stories, the patient was the one requesting a treatment, in the other nine cases the relatives (surrogate decision-makers such as spouses or children) were requesting further treatment. Six patients were aged 25-45 years, three were aged between 45 and 65 years and five were 65 years or older. Physicians explained their choice of patient by different reasons, either the young age of the patient, the acuteness of the situation or the frustration they felt when a treatment was requested or even demanded they had not proposed and felt was unnecessary, futile or even harming the patient.*

Table 2 (page 21): *End-of-life situation 14*

Discussion (page 18, lines 19-22): *All physicians described a case that had had in impact on them, for instance because of the young age of the patient. Interestingly, this is not described as a mechanism of driving overtreatment. However, it was mentioned by several physicians that age did play a role in the perception of the family that all should be done to save the life of the patient .*

To this end, the title of the manuscript should also be revised to clarify that the manuscript refers to a situation where patients/family requested treatment vs. the physician's advice (i.e. it is not referring to any overtreatment in general).

We appreciate this suggestion and have altered the title.

Title: *Requests for futile treatments: what mechanisms play a role? Results of a qualitative study among Dutch physicians*

Also, it is perhaps confusing as to why the treatment was continued against the advice of the physician; if the benefits of treatment were not felt to outweigh the risks, then why were the treatments offered? Can the authors offer at least one-two examples? (e.g. presumably something like continuing ventilatory support or tube feeds despite likelihood of benefit?)

We have not been clear enough about the fact that the requested treatments were not offered by the physician, but requested by patient or relatives on their own account. We have altered the methods section.

Methods (page 8, lines 8-13): *Physicians were asked to tell about a case from their own personal experience, in which the patient or family requested continuing or starting of treatment which were not offered or advised by the physician. No limitations were given regarding type of underlying disease, patients' age, sex or cultural background. After narrating the patient story, the fifteen mechanisms described by the SCoAEoLC (see table 1) were presented and the interviewer asked them whether the mechanism played a role in that case.*

The authors explain that some of the key findings were related to challenges that physicians felt when addressing patients' relatives. This suggests that it may have been superior to stratify results by those when physicians felt overtreatment was requested directly by a patient and when overtreatment was requested by a relative(s). Further, it may be helpful to at least understand the cases in which a larger group of relatives were involved, or to distinguish between cases where a relative was a designated health care proxy / surrogate decision-maker.

This is indeed a valid observation, and one that requires some more explanation. We have tried to illustrate this in table 2 (patient communicator or family communicator) but agree with the reviewer that this is not explicit enough. A designated health care proxy was never described. The relatives that requested treatment were the surrogate decision-maker (in two cases the spouse, in one case the sister and in the remaining six cases more than one child or a combination of family-members). We have altered the result section.

Results (page 10, lines 2-4): *All physicians described a patient facing an end-of-life situation. In five patient stories, the patient was the one requesting a treatment, in the other nine cases the relatives (surrogate decision-makers such as spouses or children) were requesting further treatment.*

Table 2 (page 21) was also altered:

Treatment requested by patient	5
---

Treatment requested by one relative	3
Treatment requested by more than one relative	6

Can the authors provide the semi-structured interview templates for review?
 We will add the semi-structured interview template as a supplementary file.

Specifically, were there specific aspects of the case that physicians were asked to recollect and discuss during the semi-structured interview?

Physicians were not asked to recollect and discuss specific aspects of the case.

The authors make several important points, and which ideally would be referenced with citations. For example, physicians finding it difficult to talk about death (page 10 of 26, line 19). Some of these points (including page 10 of 26, line 19) would likely fit better in the Discussion (e.g. page 11 of 26, lines 3-8; page 12 of 26, line 11).

We agree with the reviewer parts of the result section better fit in the discussion section. As a result, we have removed these points from the result section, added them to the discussion and have included citations were needed.

Results (page 11, lines 12-14): *The tendency to 'not give up' was considered to be a factor in thirteen out of the fourteen described cases. Interestingly, the difficulties of talking about death and end-of-life topics in general, were in most cases attributed to the patient and family.*

Results (page 12, lines 9-11): *In our interviews, we found this mechanism was recognized as being an important factor in almost all cases. In thirteen cases physicians described that the patient or family did not want to give up, even when the physician had told them further treatment would be futile.*

Results (page 13, lines 9-10): *Physicians described, sometimes in detail, their unfamiliarity with cultural differences between themselves and (family of) their patients.*

Results (page 13, lines 16-17): *In our interviews we found physicians used the medical perspective to convince patients not to continue treatment.*

Discussion (page 15, line 8 – page 16, line 6): *The three mechanisms that were mentioned most as contributing to overtreatment are closely intertwined. Patients are often hesitant to discuss the approaching end of life and to accept the fact that they are going to die. This reluctance to talk about the end of life is sometimes enhanced by relatives, who can pressure patients not to give up. To talk about death when the patient and family are still aiming for a cure is difficult. Furthermore, in certain cultures it is not customary to discuss the diagnosis, or the approaching death with the patient, making it even harder for the physician to find out what, in a certain case, appropriate care is. As patients and relatives find it difficult to discuss and accept approaching death, physicians often find themselves in a position where patients preferences and values are unknown to them, due to the impossibility to communicate directly with the patient in the acute setting of an end-of-life situation. The general tendency in society is 'to not give up' in the 'fight' against a certain disease. Dealing with a disease is often seen as a 'fight' or as a 'war'.(13) When dealing with cancer, for instance, patients are supposed to 'fight' the disease. If they are cured they are seen as 'winners' who have 'conquered' the disease. These war-like metaphors are seen also in funding campaigns, such as 'the war on cancer'.*

The cultural and spiritual background of patients can be contributing factors in the perception of physicians that the patient is being subjected to overtreatment. For instance, in some cultures it is not common practice to inform the patients of diagnosis and prognosis. Patients and relatives can also hold specific views on the meaning of suffering and whether pain medication is indicated: 'suffering is purification'. These views can bring patients in a situation that physicians find difficult to accept professionally.

Results (page 16, line 21 – page 17, line 4): *The SCoAEoLC considered one of the drivers of overtreatment to be the idea that physicians take too much of a 'medical' perspective, meaning they focus too much at the medical side of a disease, foregoing other important aspects, such as wellbeing of the patients, or social and cultural aspects. According to the SCoAEoLC this means the 'medical perspective often dominates the decision-making process, even when multidisciplinary consultation (MDO) is involved. Such bodies often consist only of medical specialists, which can overshadow reflection on social, mental, spiritual, cultural and ideological aspects, as well as general well-being'.(11) However, in our interviews, we found this driver needs to be interpreted in a more nuanced way.*

The authors state in the first sentence of the Discussion (page 14 of 26) that “not all mechanisms were considered equally important.” However, were they evaluating the importance of the mechanisms, more their frequency of mention as well? How were physicians asked to rank the importance of these mechanisms? Similarly, in the conclusions (page 17 of 26, line 8) the authors refer to “the major driving factors” but it is not clear exactly what “major” refers to and the methods should clarify this topic.

We agree with the reviewer that the subjective part of this part of the manuscript is not the best way to describe our research. We have not asked to rank the different mechanisms, as this is qualitative research, and quantitative results are not aimed for. We have therefore removed the more subjective parts of the discussion and instead have used objective terms. This was changed throughout the manuscript.

Abstract (page 2, line 20 – page 3, line 2): *From these interviews, it was found that three closely related mechanisms were mentioned most as being a driver of overtreatment, as perceived by the physician: ‘death is not a common topic of conversation’, ‘never give up’ is the default attitude in our society’ and ‘patients’ culture and outlook on life influences their perception of death’.*

Abstract (page 3, lines 5-10): *Of the fifteen mechanisms described by the SCoAEoLC, not all mechanisms were mentioned as driving overtreatment. Three mechanisms were mentioned most as being a driver of overtreatment (‘death is not a common topic of conversation’, ‘never give up’ is the default attitude in our society’ and ‘patients’ culture and outlook on life influences their perception of death’), some played no role at all, and others were considered to be inhibitors of overtreatment, especially the mechanism ‘medical view taking priority’.*

Discussion (page 14, lines 2-7): *In the present study we found that of the fifteen mechanisms described by the SCoAEoLC not all mechanisms were considered important in driving overtreatment. Three mechanisms were mentioned most as being the drivers of overtreatment (‘death is not a common topic of conversation’, ‘never give up’ is the default attitude in our society’ and ‘patients’ culture and outlook on life influences their perception of death’), some played no role at all, and others were even considered to be inhibitors of overtreatment, especially the mechanism ‘medical view taking priority’.*

Conclusion (page 19, lines 2-5): *We have found that three mechanisms were identified most often as a driving factor in overtreatment: ‘death is not a common topic of conversation’, ‘never give up’ is the default attitude in our society’ and ‘the great unknown: patients’ culture and outlook on life influences their perception of death’.*

Throughout the paper (including in the Title), It may be more accurate to refer to citation 11 by its name instead of only referring to the last name of the first author/Chair of the Steering Committee. We agree with this comment. All throughout the manuscript, the name “Van der Wal: has been changed to *the SCoAEoLC*, with explanation of the abbreviation the first time the report was mentioned.

Page 2 of 26, line 30: “addressed this problem” should be “attempted to address this problem” This sentence has been altered according to the suggestion made by the reviewer.

Abstract (page 2, line 8): *In 2015 the Royal Dutch Medical Association attempted to address this problem and distinguished several mechanisms that were thought to drive overtreatment*

Page 3 of 26: Not all of the items in the Conclusions are mentioned in the Results of the abstract We thank the reviewer for this comment. A sentence has been added to the results of the abstract. Abstract (page 3, lines 2-3): *The mechanism ‘medical view taking priority’ was mentioned to be an inhibitor of overtreatment.*

Page 6 of 26, line 55: It is not just that some patients will accept even a small chance of success, but also be willing to accept risks (i.e. medical decision-making is often based on a balance of benefits and risks – not solely on the possibility of benefit)

We agree with the reviewer. Thus, we have adjusted the sentence

Introduction (page 6, lines 21-23): *Some patients will request as much treatment as possible and accept even a small chance of success and are willing to accept risks, while other patients will be much more reluctant in accepting interventions*

Page 11 of 26, line 54: It is not clear what aspect of this quotation from interview 15.1 relates to “easy to cure” problems.

We thank the reviewer for this valuable comment. As a result, we have scrutinized all quotes and changed some of them. Furthermore, we have revised the accompanying numbers to make the manuscript more concise.

Quote (page 11, lines 17-18): *“The family was not used to talk about the end of life, no. But they found it also hard to talk about bad news in general.”* (interview 10)

Quote (page 11, line 22): *“Actually, we talked about it over and over again.”* (interview 9)

Quote (page 12, lines 1-2): *“Only because she refused to talk about it, not because we were not willing to discuss the subject.”* (interview 15)

Quote (page 12, lines 5-6): *“So, in this case, while I do possess the skills to talk about it, I found it extremely difficult”*(interview 8)

Quote (page 12, line 7): *“It caused frustration on my part, and also a feeling of helplessness.”*(interview 13)

Page 14 of 26, lines 34-39: Is it that patient preferences and values are unknown as the authors state, or is it that they are not elicited by physicians? It seems that the latter is also important. Indeed it is the case that patients preferences and values are unknown to the physician, because in a large number of the presented cases it was not possible to communicate with the patient. We have tried to clarify this by adding a section to this sentence

Discussion (page 15, lines 14-18): *As patients and relatives find it difficult to discuss and accept approaching death, physicians often find themselves in a position where patients preferences and values are unknown to them, due to the impossibility to communicate directly with the patient in the acute setting of an end-of-life situation. This was the case in the majority of the narrated patient stories.*

Lines 44-46: Is it that the patients or relatives considered the proposed intervention to be overtreatment before the decision or afterwards?

This is a very interesting question, but not one we can answer based on our research. We have not interviewed the patients or relatives.

Lines 48-51: The question that the authors pose here likely varies across clinical contexts.

Additionally, the physicians may consider an intervention to be overtreatment – but another question remains as to why the physicians are offering the intervention to patients. If it is truly overtreatment and benefits do not outweigh risks (or meets one of the other definitions stated in the Introduction’s paragraph starting at line 30), this should have been further explored. While the authors allude to a potential explanation on page 15 (if the patient asks for an intervention then physicians may be most likely to go with it), it is still likely that physicians would have introduced the potential intervention to the patient in most cases.

We thank the reviewer for this observation. We have not been clear enough about the fact that the requested treatments were not offered by the physician, but requested by patient or relatives on their own account. We therefore have added a sentence to the methods section.

Methods (page 8, lines 8-13): *Physicians were asked to tell about a case from their own personal experience, in which the patient or family requested continuing or starting of treatment which was not offered or advised by the physician. No limitations were given regarding type of underlying disease, patients’ age, sex or cultural background. After narrating the patient story, the fifteen mechanisms described by the SCoAEoLC (see table 1) were presented and the interviewer asked them whether the mechanism played a role in that case.*

Pages 21 and 22 of 26: The results of Figures 1 and 2 should be discussed in further detail in the paper

We have added further information about both figures in the result section.

Results (page 10, line 22 – page 11, line 3): *When attributing the perceived drivers of overtreatment to the different parties, there was a distinct difference between the mechanisms. As is shown in figure 1, mechanism no. 1 ‘death is not a common topic of conversation’ was in all cases attributed to the patient or family, whereas mechanism no. 14 ‘the great unknown: patients’ culture and outlook on life influences their perception of death’ was assigned to the physicians themselves. Mechanism no. 2 ‘never give up’ is the default attitude in our society’ was attributed mainly to patients and family, with two cases assigning this mechanism to the physician or a combination of parties.*

Results (page 11, lines 6-7): *As is shown in figure 2, this mechanism was in all cases attributed to the physician.*

Reviewer: 2

Reviewer Name: Loai Albarqouni

Institution and Country: Bond University, Australia Please state any competing interests or state 'None declared': None declared

Thanks for inviting me to review this manuscript. The authors in this study aimed to investigate the drivers of overtreatment among 14 Dutch physicians and found that 3 mechanisms played an important role as the main drivers of overtreatment ('death is not a common topic of conversation', 'never give up' is the default attitude in our society' and 'patients' culture and outlook on life influences their perception of death'). The manuscript is well written and easy to follow. Few comments to strengthen the manuscript:

Introduction:

- Need to be revised to make it clear, concise, and to the point connecting the rationale for this research project to the objectives.

We would like to thank the reviewer for his valuable time and effort and the suggestions and comments. We have revised our manuscript accordingly and feel the content has improved.

- Van der Wal mechanisms are not very common, therefore, I wonder whether to include the name in the title?

We agree with this comment. Not only have we changed the title, but all throughout the manuscript, the name "Van der Wal" has been changed to *the SCoAEoLC*, with explanation of the abbreviation the first time the report was mentioned.

Title: *Requests for unnecessary treatments: what mechanisms play a role?*

Methods

- Selection/sampling process need to be adequately reported. For instance, author mentioned the 14 were randomly selected by without further details.

We thank the reviewer for this comment. We have added several sentences to the method sections Methods (page 7 line 21 – page 8 line 4): *We purposively sampled eighteen physicians differing in years of working experience and medical specialty to participate. Names were randomly selected from the different departments of (academic) hospitals or groups of local general practitioners by using the accessibility guide of the Erasmus MC as well as the list of General Practitioners in the region. Fourteen physicians from five different medical areas (internal medicine, general practice (GP), intensive care, surgery, and oncology) agreed to participate. Four physicians were interested in the topic, but were not able to find sufficient time in their schedule to participate. Their work experience ranged from one to thirty-five years. All interviews were conducted between March 2014 and November 2015.*

VERSION 2 – REVIEW

REVIEWER	Sanket Dhruva UCSF School of Medicine
REVIEW RETURNED	02-Mar-2020

GENERAL COMMENTS	The authors have performed an impressive and excellent revision of their manuscript. I have just a few minor comments: 1) It would be helpful to state explicitly in the Discussion that these findings relate to generally younger individuals (compared to average life expectancy) and end-of-life decision-making. Of note, in the Response Letter, the authors stated that 13 of 14 cases related to end-of-life care, but the first sentence of the Results of the manuscript states that "All physicians described a patient
---

	facing an end-of-life situation,” which is confirmed in Table 2 (14 of 14 patients). 2) Given that all of the cases relate to end-of-life care, this could be incorporated into the title of the manuscript. One suggestion is, “Requests for futile treatments at the end-of-life: what mechanisms play a role? Results of a qualitative study among Dutch physicians.” 3) I am not sure that this sentence in the “Article Summary” is exactly accurate, “A wide variety of patients was described, making the findings applicable to several types of situations and/or physicians” because all of these patients were near the end-of-life. I suggest that this should be revised to more accurately reflect the patient cases (although I recognize the overall goal of the study was to examine overuse more broadly).
--	---

VERSION 2 – AUTHOR RESPONSE

Reviewer(s)' Comments to Author:

Reviewer: 1

Reviewer Name: Sanket Dhruva

Institution and Country: UCSF School of Medicine, USA Please state any competing interests or state 'None declared': None declared.

The authors have performed an impressive and excellent revision of their manuscript. I have just a few minor comments:

1) It would be helpful to state explicitly in the Discussion that these findings relate to generally younger individuals (compared to average life expectancy) and end-of-life decision-making. Of note, in the Response Letter, the authors stated that 13 of 14 cases related to end-of-life care, but the first sentence of the Results of the manuscript states that “All physicians described a patient facing an end-of-life situation,” which is confirmed in Table 2 (14 of 14 patients).

We would like to thank this reviewer for this observation and suggestion. The text in the manuscript is correct, all 14 patient cases were related to end-of-life care. We have stated more explicitly that these findings relate to generally younger individuals and end-of-life decision making.

All physicians described a case that had an impact on them. The described patients were generally younger individuals and the decisions to be made were related to end-of-life decision making.

2) Given that all of the cases relate to end-of-life care, this could be incorporated into the title of the manuscript. One suggestion is, "Requests for futile treatments at the end-of-life: what mechanisms play a role? Results of a qualitative study among Dutch physicians."

We very much appreciate this suggestion and have changed our title.

Requests for futile treatments at the end-of-life: what mechanisms play a role? Results of a qualitative study among Dutch physicians.

3) I am not sure that this sentence in the "Article Summary" is exactly accurate, "A wide variety of patients was described, making the findings applicable to several types of situations and/or physicians" because all of these patients were near the end-of-life. I suggest that this should be revised to more accurately reflect the patient cases (although I recognize the overall goal of the study was to examine overuse more broadly).

We agree with the reviewer and have changed this sentence:

- A wide variety of patients facing end-of-life decisions was described, making the findings applicable to several types of situations and/or physicians